# Aquaphotomics Monitoring of Lettuce Freshness during Cold Storage

**DOI:** 10.3390/foods12020258

**Published:** 2023-01-06

**Authors:** Flora Vitalis, Jelena Muncan, Sukritta Anantawittayanon, Zoltan Kovacs, Roumiana Tsenkova

**Affiliations:** 1Department of Measurements and Process Control, Institute of Food Science and Technology, Hungarian University of Agriculture and Life Sciences, Somlói Street 14-16, H-1118 Budapest, Hungary; 2Aquaphotomics Research Department, Graduate School of Agricultural Science, Kobe University, 1-1, Rokkodai, Nada, Kobe 657-8501, Japan

**Keywords:** lettuce, cold storage, water, aquaphotomics, near-infrared spectroscopy, water molecular structure, monitoring, non-destructive measurements, freshness, shelf life

## Abstract

Fresh-cut leafy vegetables are one of the most perishable products because they readily deteriorate in quality even during cold storage and have a relatively short shelf life. Since these products are in high demand, methods for rigorous quality control and estimation of freshness that are rapid and non-destructive would be highly desirable. The objective of the present research was to develop a rapid, non-destructive near-infrared spectroscopy (NIRS)-based method for the evaluation of changes during cold storage of lettuce using an aquaphotomics approach to monitor the water molecular structure in lettuce leaves. The reference measurements showed that after 6 days of dark, cold storage, the weight and water activity of lettuce leaves decreased and β-carotene decreased, while chlorophylls slightly increased. Aquaphotomics characterization showed large differences in the lettuce leaves’ spectra depending on their growth zone. Difference spectra, principal component analysis (PCA) and linear discriminant analysis (LDA) confirmed the differences in the inner and outer leaves and revealed that spectra change as a function of storage time. Partial least squares regression (PLSR) allowed the prediction of the time spent in storage with a coefficient of determination of R^2^ = 0.80 and standard error of RMSE = 0.77 days for inner, and R^2^ = 0.86 and RMSE = 0.66 days for outer leaves, respectively. The following water absorbance bands were found to provide the most information in the spectra: 1348, 1360, 1373, 1385, 1391, 1410, 1416, 1422, 1441, 1447, 1453, 1466, 1472, 1490, 1503, 1515, 1521, 1534 and 1571 nm. They were further used as water matrix coordinates (WAMACs) to define the water spectral patterns (WASPs) of lettuce leaves. The WASPs of leaves served to succinctly describe the state of lettuces during storage. The changes in WASPs during storage reveled moisture loss, damage to cell walls and expulsion of intracellular water, as well as loss of free and weakly hydrogen-bonded water, all leading to a loss of juiciness. The WASPs also showed that damage stimulated the defense mechanisms and production of vitamin C. The leaves at the end of the storage period were characterized by water strongly bound to collapsed structural elements of leaf tissues, mainly cellulose, leading to a loss of firmness that was more pronounced in the outer leaves. All of this information was reflected in the changes of absorbance in the identified WAMACs, showing that the water molecular structure of lettuce leaves accurately reflects the state of the lettuce during storage and that WASPs can be used as a multidimensional biomarker to monitor changes during storage.

## 1. Introduction

Lettuce (*Lactuca sativa* L.) is the most important among leafy vegetables and is usually eaten raw [1]. As it is a cold season crop, it is widely grown in temperate and subtropical regions. About two thirds of the worldwide production area is in Asia [2]. In Japan, it is the fourth most frequently used leafy vegetable in the fresh-cut food market as it is one of the main ingredients in packaged “ready-to-eat” salads [3]. A very wide range of varieties can be found on the shelves of markets including crisphead (iceberg), romaine, butterhead and curled (stem) lettuce. The color of the leaves varies from deep green to red, depending on the variety [4]. Lettuce is characterized as being low in calories and fat, with water content being about 95%; however, it is a good source of fiber, vitamins, minerals and many other bioactive compounds (folate, carotenoids, phenolic compounds…) with beneficial health effects. In general, the darker the leaf color, the higher the nutritional value of the leaves [1,5]. The leaves are the primary site of photosynthesis in plants which generally makes them the most nutrient-dense and most perishable part of the vegetable. 

Due to high consumer demand, lettuce that is offered fresh usually undergoes various pre-processing steps, even during minimal processing, aiming to extend the shelf life of salads as much as possible. In the case of minimally processed lettuce, the heads should be harvested with special care due to the mechanical and physiological fragility of the leaves. Then, the lettuce should be trimmed manually to remove discolored, low-quality, decayed leaves. In some cases, gentle washing is also applied to reduce mechanical and microbiological contamination. This is followed by packaging and storage before distribution [4]. Generally, harvest maturity and post-harvest technologies fundamentally determine the shelf life of lettuces, which show additional metabolic activity depending on environmental conditions resulting in various physicochemical changes [6]; hence, finding the optimal storage conditions is of the utmost importance [7]. 

Depending on the growth zone of the leaves (inner or outer), the physical, chemical and thus the organoleptic properties are different, due to differences in exposure to the environment. Furthermore, they react differently to the environmental impacts they are exposed to. Aguero et al. have exhaustively studied inner, middle and outer leaves, and found that, compared to the inner leaves, the outer leaves show more intense changes in color parameters, chlorophyll content and the levels of total soluble solids (TSS) [8]. It was also observed during sensory evaluation that the inner leaves of lettuce declined slower than that of outer leaves; this can be attributed to the fact that the outer leaves may function as barriers to oxidation and dehydration. Similar conclusions were drawn by Baslam et al. who studied three different types of lettuce [9]. They noticed an accumulation of soluble sugars in the inner leaves and explained that inner leaves act as a strong sink of sugars synthesized in the outer leaves in order to supply the youngest leaves with carbohydrates for metabolic purposes and support for growth. 

To the best of the authors’ knowledge, currently there are no non-destructive measurement methods that can objectively and fully describe the physiological status of leafy vegetables during storage. Due to the easy leaf damage that is typical of lettuces, non-destructive and even non-contact methods are a highly desirable solution that could allow changes during storage to be monitored using digital fingerprints. Near-infrared (NIR) spectroscopy has a long history of use for similar purposes in the horticultural and food sector [10,11,12,13] and present an excellent candidate for this purpose. There have also been previous instances of the use of NIR spectroscopy for the estimation of storage time/post-harvest age/freshness of lettuce [14,15,16,17]. Although the studies reported success in the estimation of storage time, only one utilized the informational value of NIR spectra to provide an explanation of the measurement basis and attempted a physiological interpretation of the achieved results by connecting the NIR-based prediction of storage time with some physiologically important changes, such as pigment degradation [18]. However, the NIR spectra can provide much more information about the degradation process and also provide new ideas on how it can be influenced in order to maintain freshness and prolong the shelf life, as some recent studies have shown [19]. There is also very little information, in general, about the spectral features of lettuce during storage and which would be the most suitable tissues for freshness evaluation, a very important aspect as this was shown for the estimation of freshness of cabbage [20]. Finally, despite the high water content and many emerging studies connecting the role of water molecular structure with the preservation of fresh fruits, vegetables and grain [21,22,23,24,25], the state of water in lettuce and how it changes during post-harvest storage is not a very well-researched area. 

Traditional NIR spectroscopy may be limited by the high water content, which is a given for lettuce, but the modern concept of aquaphotomics has opened up a new dynamic, non-invasive way of biosystem monitoring based on spectral measurements of the water molecular matrix [26]. Lettuces are characterized by a high water content, which accounts for approximately 95% of their total mass [2,27]; the other components are present in negligible quantities. Consequently, the changes in lettuces during storage are not only due to quantitative changes, but also qualitative changes in water, in other words the reorganization of the water molecular network in the tissues of lettuce leaves. 

Using aquaphotomics analysis and interpretation [28], the mechanism of maintaining freshness and how this is influenced by cooling technology or other innovative preservation strategies can be better understood. Such experience and knowledge could serve as a basis for improved refrigeration, optimization, maintaining freshness and prolonged shelf life [24]. 

Motivated by these reasons, this research was performed with the following objectives: (1) development of a non-destructive methodology for monitoring changes during the storage of lettuce; (2) identification of distinctive water absorbance bands in the near-infrared spectrum which can be used as measurement hubs—i.e., water matrix coordinates (WAMACs) [26] that carry information about the state of the lettuce; (3) development of a novel marker (biomarker) based on a water spectral pattern (WASP) [26] defined by identified WAMACs for the description of the status of the water molecular structure in the inner and outer zones of lettuce and 4) characterization of changes in the inner and outer zones of lettuce as a function of time spent in cold storage using WASPs.

## 2. Materials and Methods

### 2.1. Materials and Experimental Conditions

Ten heads of lettuce (JA Topia-Hamamatsu, Hamamatsu, Japan) were purchased in a local supermarket, ensuring the vegetables were without any damage or defects. After being transported to the laboratory, the lettuce packaging was removed and they were stored “as they were” without washing or any other pre-conditioning operation in the commercially available refrigerator in dark conditions (model no. YRC-080RM2, NICHIEI INTEC CO., LTD, Tokyo, Japan) and monitored for 6 days. The operating temperature of the refrigerator was set to 0–2 °C while the relative humidity was 91–95%. Five lettuce heads were used for spectral analysis, while the other five were used for weight measurements and changes in pigments in the outer leaves. 

### 2.2. Methods

#### 2.2.1. Weight Measurements

Five lettuce heads were unpacked, 1 outer leaf was separated from each lettuce head on the first day of monitoring and their weight was measured daily during the six days of storage using an analytical balance with 0.0001 g of precision (model AUX220, Shimadzu, Kyoto, Japan). The duration of storage was selected to be six days because this is the usual time limit for the stored produces to be sold [29]. On the last day of monitoring, 2 outer leaves were separated after the weight measurement. 

#### 2.2.2. Water Activity Measurements

Water activity [30,31] (aw) measurement was used as a reference method to obtain information about the freshness and microbiological stability of lettuce heads. The water activity of food is commonly determined by measuring the vapor pressure of the headspace in a closed contained area after equilibrium is established between the headspace and the sample [32]. The measurements were conducted on the inner and outer lettuce leaves once a day during the six days of storage. To perform the measurements, an Aqualab 4TE aw meter (Decagon Devices, Inc., Pullman, WA, USA) was used. The instrument is equipped with a dew point sensor which enables the direct measurement of water activity and provides an accuracy of ±0.003. 

#### 2.2.3. Evaluation of Color Changes

Visible spectroscopy is routinely used for non-destructive assessments of pigments in fruits and vegetables based on their spectral features [33]. To detect color changes during storage, only the outer leaves of lettuces were analyzed according to their VIS-NIR spectra using an XDS-RLA benchtop spectrometer (FOSS, Denmark) on the first and last day of storage. The spectral data were recorded in the wavelength range of 400–700 nm, with a resolution of 0.5 nm. For the measurements, the samples were prepared by excising a rectangle of green tissue from the leaves approximately 2 × 3 cm in size, using a scalpel. The resected sample was placed directly into the sample holder and fixed into a stable position using a space adjuster. The rectangular lettuce leaf pieces were cut from 2 different leaves. During measurements, 10 consecutive spectra per lettuce sample were recorded. 

#### 2.2.4. NIR Spectral Acquisition

NIR spectroscopy for non-destructive quantitative and qualitative assessments of fresh fruits and vegetables has a long history of applications and is a well-developed methodology [33]. Spectral measurements of five lettuce heads were performed using the portable instrument MicroNIR (Viavi Solutions, formerly JDSU, USA) in diffuse reflectance mode. The spectra were recorded in the wavelength range of 908–1670 nm at three measurement positions, and at each position 5 consecutive spectra were acquired. The measurements were performed at the same time, at 3.00 PM every day. For the measurements of inner leaves, the outer leaves were gently opened to allow the positioning of the instrument on the exposed surface of the inner leaves. A total of 900 spectra were collected during storage (5 lettuce heads × 6 days × 6 positions × 5 consecutive scans). 

### 2.3. Data Analysis

#### 2.3.1. Statistical Analysis of Weight, Water Activity and Pigment Changes

The mean and standard deviation (SD) of weight and water activity for 5 lettuce heads were calculated and graphically presented as a function of storage time using Origin Pro 2018 software (OriginLab Corp. Northampton, MA, USA).

The spectra of lettuce leaves acquired on the first and last day of storage were trimmed to only the visible region of 400–700 nm, and raw and standard normal variate (SNV) [34] transformed spectra were inspected for the absorbance bands attributable to pigments. The values of absorbance at the identified absorbance bands were compared using the paired samples t-test to evaluate the differences in pigments between the first and the last day of storage of lettuce leaves. The pre-processing of spectra was performed using commercially available software Pirouette v4.5 (Infometrix Inc, MA, USA), while visualization and statistical analysis were performed using Origin Pro 2018 software (OriginLab Corp. Northampton, MA, USA).

#### 2.3.2. Aquaphotomics Multivariate Data Analysis

Multivariate analysis was performed using R-project v 3.6.3 and aquap2 package [35], following the protocol for the analysis as given in a previous publication [28]. The analysis was performed separately for the spectral data collected from the outer and inner leaves of the lettuce. The spectra were evaluated in the wavelength range of 1300–1600 nm, the first overtone of water, commonly used in aquaphotomics research studies. 

The data evaluation started with the inspection of the difference spectra which were calculated by subtracting the averaged spectrum of Savitzky–Golay-smoothed (second order polynomial, 11 points) [36] and standard normal variate (SNV) transformed spectra [34] collected on the first storage day from the spectra collected on other storage days pre-treated in the same way. To identify wavelengths covering notable absorbance changes caused by the cold storage, the major peaks of the difference spectra were investigated. 

Data pre-treated in different ways, then subjected to principal component analysis (PCA) [37], were used as inputs for linear discriminant analysis (LDA) [38]. 

PCA is a well-known chemometric technique widely used for a reduction in the total number of variables in multivariate data analysis [37]. PCA is used for constructing the new dimensions of a dataset into a smaller one, but retaining most of the original data variability in a much-reduced space. This new space is defined by so-called principal components (PCs) which represent the linear combinations of the original variables, orthogonal to each other (uncorrelated), and which contain the maximum variance within them. The output of PCA consists of a loadings matrix, representing the PCs, and score matrix, which provides the coordinates of the original spectra in the space defined by PCs. PCA was used to investigate the spectra of lettuce, eliminate outliers and examine the existing patterns and trends. 

LDA is a common supervised classification technique which requires a priori knowledge of sample groups (classes) [38]. It is similar to PCA in terms of reducing the dimensionality of data, but LDA maximizes the ratio of between-class variance to within-the-class variance and thereby gives maxima separation between classes [39]. In this work, LDA was used with a purpose of detecting differences in samples and classifying them according to storage time. The optimal number of principal components (NrPCs) to be used for the modeling was determined by collecting and comparing classification results up to 30 NrPCs during threefold cross-validation. The NrPCs providing the highest classification accuracy during validation and, at the same time, the lowest difference between average correct calibration and validation accuracies, were used in the actual LDA modeling. The classification models were tested using “leave-one-lettuce-out” validation (LOLO), meaning that model calibration was performed on the data of four lettuce heads; then, data of the omitted lettuce head were projected into the calibration model during validation. The procedure was repeated five times (leaving out another dataset of a sample replicate for validation in each round) to ensure that the data of each lettuce had been involved in the model construction or validation once; then, the average of the classification accuracies was calculated. Meanwhile, overall wavelength contribution was also calculated and plotted which enabled the identification of descriptive bands relating to the spectral changes in lettuce. 

Partial least squares regression (PLSR) is a well-known, quantitative, supervised method of analysis that describes the behavior of the dependent variables as a function of the independent variables (spectra) using data compression to reduce a large number of measured collinear spectral variables to a few orthogonal latent variables (LVs) that describe the maximum covariance between independent variables and dependent variables [40]. PLSR, in this work, was applied as a qualitative analysis to discover the relationship between the progression of storage time and the spectral changes [41]. Similarly to PCA–LDA, each predictive model was tested using “leave-one-lettuce-out” validation (LOLO) when the dataset corresponding to a lettuce was excluded from the calibration set, and the built model was validated with the previously omitted dataset (CV). The procedure was repeated until each sample replicate was involved in the model calibration and validation set. The accuracy of PLSR models was determined based on the coefficient of determination (R^2^) and root mean square error (RMSE) calculated during model calibration and validation [42]. The number of latent variables (NrLV) used in the modeling was the one with the lowest RMSE_CV_ value. In order to identify absorbance bands particularly affected by cold storage, the regression vectors of PLSR models were further evaluated separately for inner and outer lettuce leaves. The location of peaks, i.e., main contributing variables (wavelengths), were determined following the steps as described in a recent publication [43]. 

To investigate the structural alterations in the water molecular network of lettuce leaves during storage, the absorbance patterns at representative water absorbance bands were further analyzed. In order to find the most informative absorbance bands, the major peaks found in the difference spectra, PCA loadings, LDA wavelength contributions and PLS regression vectors were summarized and compared. The aim of this analysis was to discover and select consistently repeating absorbance bands among the most influential variables. These absorbance bands were afterwards used as water matrix coordinates (WAMACs—absorbance bands attributed to specific water molecular conformations), with their combination defining the water spectral pattern (WASP). WASP is, therefore, an integrative marker that in a succinct way describes the state of lettuce leaves and time dynamics during storage. The differences at these specific wavelengths (WAMACs) compared to the first storage day were calculated and visualized in radar plot-like aquagrams [28,44], which present average normalized absorbance values of smoothed (Savitzky–Golay 2nd-order polynomial filter with 17 points) and SNV-pre-treated data recorded on each day of storage. The WASPs defined by these bands have helped to elucidate the complex changes that occurred during the cold storage of lettuce. 

## 3. Results and Discussion

### 3.1. Weight Change

Variations in the initial phase of storage can be mainly related to the water content and its phase changes (liquid to gas). Changes in the average fresh weight of lettuces during the six days of storage are presented in Figure 1. A fairly large deviation was observed in the results which can be attributed to the naturally high variability in the sample replicates, despite the fact that they were of the same variety and origin and were handled in the same way. The average weight of freshly purchased samples was the highest on the first day. Interestingly, by the third day of storage, the average weight had dropped considerably. On the fourth and fifth day of storage, similar values were recorded compared to the second day, which decreased slightly by Day 6. These differences may have been caused by respiration, transpiration, natural deterioration of lettuces and microbial activity. The weight change curve during the six days of storage is in agreement with previously published research results by Vargas-Arcila et al., in which they monitored four lettuce varieties (Graziella, Lollo Rossa, Paris Island, Alpha) during 12 days of storage at 5.5 °C (90% RH). Their results obtained during the determination of water content in the Alpha variety are similar to the weight change presented here. Regarding the tendencies, the weight decreases linearly up to about half of the storage period and then increases again slightly. In our case, the weight on the last day had decreased by about 11% compared to the weight measured on the first day of storage. 

This behavior is most typical for poor water retainer loose leaf lettuce varieties [6] that are particularly sensitive to excessive environmental changes (e.g., temperature and light) causing the wilting of the plant [45]. These types include green leaf, red leaf, butterhead and romaine lettuce commonly used in salads, that grow leaves from a central stalk and do not form a compact head [45,46], which is also the case with the variety of lettuce used in our study. 

### 3.2. Water Activity 

Water activity, which is one of the most important measures of food stability, is defined as the ratio of the vapor pressure of water in food and the vapor pressure of pure water at the same temperature [30]. It provides information about the amount of water available to participate in the chemical reactions [47]. The activity and reproduction of microorganisms on contaminated lettuce leaves depend, among other things, on the water activity of the leaves [48]. 

Water activity was determined in the inner and outer leaves of lettuces during the six days of storage, and are summarily presented in Figure 2**.**

The water activity of lettuce appears to follow the same trend as the weight changes presented above. The average water activity values reached a minimum on the fourth storage day, followed by a gradual increase. The deviation in the results for both inner and outer leaves was dependent on the day of storage. Water activity values showed the highest variation on Day 4, suggesting a complex and different magnitude of variation between individual lettuce heads. The water activity results presented here show dependency on the location of the leaves (inner or outer). The effect of storage is more apparent and enhanced in the case of outer leaves, since they are exposed to the environment which also results in slightly higher values of a_w_ compared to the inner leaves. These findings are supported by the fact that outer leaves are also more mature than the inner ones, and catabolic processes are more intense [6]. 

### 3.3. The Changes in Pigments 

The averaged raw spectra of lettuce leaves in the visible region acquired on the first and the last day of storage are presented in Figure 3a. The spectra show a strong baseline offset which originates from physical differences in the leaves, such as differences in thickness, water content and morphological–anatomical traits of plant tissues. The three absorbance peaks could be observed at 454 nm, 479 nm and 678 nm. To better isolate the differences due to the changes in pigments, the baseline effects were removed and the averages of SNV-transformed spectra on the first and last day of lettuce storage are provided in Figure 3b.

The SNV treatment effectively removed baseline differences, but the subtle differences at the observed peaks remained. The absorbance at 454 nm can be attributed to β-carotene [49,50]. The peak at 479 nm can be attributed to the chlorophyll *b* (Chl *b*) [51], but it is also located very close to the reported absorption maxima of carotenoid zeaxanthin (478 nm) and *β*-carotene (477 nm) [52], so it is possible that this spectral feature is governed by a combined absorption of both chlorophylls and carotenoids. The 678 nm peak belongs to one of the absorption bands of chlorophyll *a* (Chl *a*) [53], corresponding to the chlorophyll absorption maximum [54,55]. 

The paired samples t-test was utilized to evaluate the differences between the absorbance measurements of the lettuce leaves on the first and last day of storage at absorbance bands 454 nm (β-carotene), 479 nm (Chl *b*) and 678 nm (Chl *a*) found to correspond to the main pigments in leaves. In all three cases, the t-test at the 0.05 level showed a significant difference between the mean values of absorbance recorded on the first and last day of storage (Figure 4a–c). 

For the 454 nm band, the mean values of absorbance on the first day of storage (mean = 1.343, SD = 0.051) and the last day of storage (mean = 1.289, SD = 0.05) were significantly different with t = 4.276 and P = 0.00008. Next, for the 479 nm band, the mean values of absorbance on the first day of storage (mean = 1.409, SD = 0.047) and the last day of storage (mean = 1.43, SD = 0.013) were significantly different with t = −2.984 and P = 0.004. Additionally, for the 678 nm band, the mean values of absorbance on the first day of storage (mean = 1.338, SD = 0.072) and the last day of storage (mean = 1.43, SD = 0.052) were significantly different with t = −5.911 and P = 0.0000003. Based on these results, it can be concluded that the content of β-carotene was significantly decreased in the outer leaves of lettuce at the end of the cold storage period, while there was a significant increase in the absorbance of chlorophylls *a* and *b*. The decrease in β-carotene was expected, as it is very sensitive to degradation, particularly oxidation [56], and its concentration usually stagnates or decreases during post-harvest storages at rates dependent on the temperature [57]. The main critical step for the loss of β-carotene is the loss of tissue integrity [58], which suggests that at the end of the storage period there is a degradation of leaf tissues in lettuce (which was consistent with the visually observed wilting of external leaves). The change in chlorophylls is usually due to degradation as an effect of aging during the storage period which leads to observable changes in color [59]. However, changes in chlorophyll content in stored vegetables depend on a series of factors such as species, variety, temperature, storage atmosphere, presence of light, etc. [29,60]. The storage temperature of 4 °C prevents a decrease in chlorophyll content [59], so the results of slightly increased chlorophylls were not entirely unexpected and are in agreement with research reports which show either no changes at 4 or 10 °C up to 12 days [29,61], or a slight increase in chlorophyll up to 8 days of storage at temperatures 4, 8 and 12 °C [62]. Slight yellowing and browning could be visually observed at the end of the storage period on the edges of the outer leaves. 

### 3.4. Aquaphotomic Multivariate Data Analysis

#### 3.4.1. Preliminary Analysis of NIR Spectral Data Difference Spectra

In this study, a handheld NIR spectrometer was employed to monitor changes occurring during the six-day-long cold storage of intact lettuce heads which allowed non-invasive and on-site investigation. The raw NIR spectra in the first overtone region of water recorded on each day of storage are shown in Figure 5a, colored according to the location of lettuce leaves in the lettuce head (inner or outer). As was expected, due to the naturally high water content of lettuce, the sample spectra were dominated by a broad absorption band in the first overtone region of water (1300–1600 nm). The highest absorption was found to be around 1450 nm, which can be assigned to the combination of antisymmetric and symmetric stretching modes of water [63] confirming that the lettuce spectra are dominated by water. The spectra of inner and outer leaves slightly overlap; however, the absorbance values of inner leaves are generally higher and the spectra show more baseline variations. The leaves in the internal and external zone of lettuce have significantly different relative water content because of the differences in the degree of tissue development which results in different tissue water holding capacity [64]. The higher spectral profile of inner leaves (which are growing, expanding leaves) in this particular region of the spectra clearly indicates a higher water content, confirming the observations that water has a key role in cell expansion and the growth of the leaves [65]. The differences between the inner and outer leaves seen even in the raw spectra are further confirmed by the results of a preliminary PCA, which showed well-separated groups of scores of inner and outer leaves on the score plots of the first few PCs (Figure 5b).

This means that the largest portion of variations in the spectral data is caused by these differences. In order to eliminate the influence of the morphological–anatomical difference between leaves and focus on the exploration of leaves’ freshness, the subsequent analysis was performed separately for inner and outer leaves. 

The average daily difference spectra (compared to the first day of storage) calculated using the data of the inner and outer lettuce leaves separately are presented in Figure 6. 

The difference spectra in general, for both leaf types, revealed increased absorbance reaching the maximum in the regions 1350–1400 nm and 1500–1600 nm, and decreased absorbance in the region 1400–1500 nm. The region between 1350–1400 nm includes absorbance bands of weakly hydrogen-bonded water: proton hydration, water solvation shells, water vapor and trapped water [24]. The increase in absorbance in this region may indicate a loss of moisture through transpiration, which is much more pronounced in the outer leaves. There are slight differences in the position of absorbance bands at which the difference in absorbance was the highest—for inner leaves the maximum difference was found around 1354–1360 nm, while for outer leaves there was a shift over time from 1379 to 1385 nm. The region between 1500–1600 nm is associated with strongly bound water molecules including crystalline and polymer-bound water [66,67,68,69]. In the case of the outer leaves, it can be observed in the difference spectra that the maximum peak in this region is being shifted towards longer wavelengths as the storage days pass, which indicates increased hydrogen bonding. The absorbance bands located at 1534 and 1565 nm are present in the difference spectra for both leaf types, but 1546 and 1571 nm are also prominent in the case of the outer leaves. The band at 1534 nm corresponds to strongly bound water [70,71] related to a water–cellulose interaction [72], that appears during the initial phases of drying [73]. The absorbance band at 1565 nm is also strongly hydrogen-bonded water [74], a feature of crystalline ice-like water [75] that increases absorbance as crystallinity increases [76]. The entire region of 1560–1570 nm was found to reflect the interaction of water with starch and sugar and is very informative with regard to the vegetative growth stage [77]. The band of 1546 nm is located very closely to the band of 1548 nm, attributed to intramolecular hydrogen-bonded OH groups in crystalline regions in a cellulose matrix of the wood that are oriented preferentially in a direction parallel to the cellulose chain, and such hydrogen bonding is strongly related to the mechanical strength [78,79]. Since, in general, bands above 1500 nm can be assigned to the first overtone of the ice-like, highly organized water molecular structures expected around hydrated macromolecules [80,81,82], it is probable that the shift observed in the position of the 1534 → 1546 → 1565 → 1571 nm in the case of outer leaves indicates a change in the degree of crystallinity and the strength of the hydrogen bonds in the structural water. 

In the case of the inner leaves, several absorbance bands emerged especially between 1400–1500 nm. Interestingly, the minima corresponding to the second storage day occurred at lower wavelengths compared to the other difference spectra. The minimum located at 1391 nm can be attributed to trapped water [83] shown to be an indicator of drying, stress, cell damage and dehydration in plants [24,84,85]. However, in subsequent days this peak shifted towards longer wavelengths (Figure 6a). This phenomenon can be explained by the fact that the inner leaves, being covered by the outer ones, are protected from external impacts, so the adaptation to the cooling conditions takes longer. It is also due to the external protection whereby, with the exception of day two, the difference spectra did not show large day-to-day differences. The absorbance bands observed at 1416, 1422, 1453 and 1472 nm can be assigned to free water molecules, hydration water, water solvation shells and molecules with three hydrogen bonds [26]. The changes in absorbance at these bands can be related to the changes in moisture content, water activity, damage due to the abiotic stress (cold storage) and viability [24]. 

In contrast to the inner leaves, the outer leaves showed impacts of the cooling process practically from the first day of storage, and the characteristic peaks appeared consistently at the same absorbance bands on each day of storage, increasing in magnitude as the storage progressed (Figure 6b). In the 1400–1500 nm range, only 1428 and 1441 nm absorbance bands were prominent. The band at 1428 nm can be attributed to hydration water [26] that may be associated with glucose molecules in cellulose, and whose absorbance intensity increases upon drying [86]. In numerous studies, this band was attributed to amorphous regions in cellulose [78,79,87,88,89,90] and an increase in absorption at this band was found to be related to an increase in density [79]. The 1441 nm absorbance band is the absorbance band of the water dimer [26], found to be possibly related to a sugar–water interaction [91,92]. 

All the relevant peaks identified here in the difference spectra of inner and outer lettuce leaves will be, together with other important absorbance bands from the subsequently performed analyses, summarized and discussed in the Section 3.4.5 Aquagrams, within the context of finding the most important bands for the description of the lettuce state during storage. With the exception of 1534 and 1565 nm, the location of the peaks did not overlap for the inner and outer leaves, but mostly fell within the ranges of WAMACs. 

#### 3.4.2. Exploratory Spectral Analysis–Principal Component Analysis (PCA)

To further investigate the effects of cold storage, PCA was applied to the raw spectral data of inner and outer leaves. Based on the PCA score plots, it was found that despite the fact that the first PCs described more than 99% of the total variance, the ordering of the scores according to storage time was most noticeable along PC 6 for inner leaves and PC 7 for outer leaves (Figure 7). The pattern of changes depending on the storage time was more pronounced and easier to observe for the scores of outer leaves indicating a stronger impact of the time spent in storage, as these leaves were without protection. 

The loadings of PC 6 for inner leaves and PC 7 for outer leaves revealed the variables’ wavelengths corresponding to the location of absorbance bands that can explain the observed pattern of changes along the storage time. Comparing the loadings of these principal components, it was found that they are nearly identical, with minor differences in magnitude in some places (Figure 8). 

A total of 22 peaks have been identified in the loadings of both inner and outer leaves’ PCA analyses. In both cases, the important wavelengths found at 1311, 1342, 1360, 1379, 1410, 1447, 1472, 1490, 1521, 1546, 1571 and 1589 nm showed positive peaks, while those found at 1317, 1348, 1373, 1391, 1422, 1466, 1478, 1503, 1534 and 1577 nm showed negative peaks. These peaks and their interpretation will be further discussed in the 3.4.5. Aquagram Section. The majority of absorbance bands identified in this way coincide with WAMAC ranges, indicating that the changes in the leaves happening during storage are changes in water molecular species, or in other words that there is a reorganization of water molecular matrix in leaf tissues during cold storage. 

The absorbance around 1310 nm can be primarily assigned to free O-H of H_2_O in small proton hydrates (H^+^(H_2_O)_5_ [93], H^+^(H_2_O)_2_ [94]). All the other absorption bands are within the ranges of the C1-C12 WAMACs and only the band of 1447 nm is slightly outside of the C8 (1448–1454 nm); however, the assignment is similar to solvation shells: the absorbance can be attributed either directly to the surface water molecules or to an envelope of hydrogen-bonded surface OH groups [67,95,96]. The absorbance bands located at wavelengths longer than 1520 nm are all related, as previously discussed, to strong, mainly polymer-bound water. These results imply that as a consequence of lettuce respiration, the free- or less-bonded water in leaf tissues (internal moisture) is being lost to evaporation, leading to the changes in the structural water, i.e., water confined by macro components which can in simpler words be described as drying and more intensive, denser packing on a cellular level. 

#### 3.4.3. Linear Discriminant Analysis of the Storage Time 

LDA was performed to classify lettuce spectra according to the day of storage and to identify wavelengths, in other words, the influential variables that have contributed the most to the detection of spectral changes, and hence the impact of storage progress. As detailed in the relevant section of “Materials and Methods”, LDA modeling and validation were performed by leaving one lettuce head out, and in this section the best possible models and results are included. LDA modeling was performed on the spectral data of inner and outer leaves separately and in both cases very accurate classification results were achieved. 

The first two linear discriminant variables (LD 1, LD 2) of calculated PCA-LDA models are reported in Figure 9. In the case of the inner leaves, the most accurate classification was achieved using the raw data (Figure 9a). The scores plot illustrates how data points are separated in the plotted discriminant space. The scores are well separated in distinctive clusters corresponding to the number of days in storage; however, a clear trend with respect to the duration of storage could not be observed. The average accuracy of correct classification during model building and validation was 99.89% and 99.56%, respectively. The number of misclassification cases was minimal and they occurred only for the second half of storage (Table 1). 

For the spectral data of outer leaves, the most accurate classification was achieved when a detrending pre-treatment was applied. In this case, the results present a slightly more overlapping pattern with each other compared to the results of the inner leaves. If an imaginary axis was drawn from the 0 point of LD 1 in the LDA score plot the results corresponding to the first–third and fourth–sixth storage days would be more clearly clustered. These LDA classification results of outer leaves demonstrate easy-to-explain differences when the classification accuracies during model building and validation were 99.78% and 96.67% (Table 2).

Another important output in defining the goodness-of-fit of classification models is the specification of key variables. The contribution of each variable (each wavelength) to the classification model is presented in Figure 10. The prominent absorbance bands identified from the discriminant analyses clearly show how the water species respond differently and in different extents to cold storage depending on the position of leaves in the lettuce head.

The common absorbance bands were found at 1336, 1366, 1422, 1466, 1497 and 1571 nm. Except for the last one, the listed wavelengths are within the ranges of C1, C2, C6, C9 and C11 WAMACs. The absorbance band at 1336 nm was a very strongly influential band in the model for SSC prediction during apple drying in two studies [97,98]. The band at 1366 nm, related to the hydration shells of ions and protons [99], was also found to be related to cellulose and water (7320 cm^−1^), and it was one of the most influential variables in models for the prediction of wood density and whose absorbance depended on the moisture conditions, being especially high for dry conditions [87,100]. It was also related to the hardness [101]. 

The band at 1422 nm is very close to the 1420 nm band, and was found to be the optimal wavelength for the prediction of vitamin C content and shrinkage of apple slices during drying [97]. Multiple studies have shown that the absorption peak of 1420 nm had a high correlation with the internal quality of fruit [102,103] and that absorption at this band generally increases as a function of storage time (similarly to the change in soluble solids content (SSC)) [103]. Another study found that the band 1420 nm was closely related to the changes during storage, specifically to loss of weight [104], while another one found it to be closely related to the moisture content of roasted coffee beans, and the most powerful wavelength to classify the defective beans [105]. Regarding the bands 1466 nm and 1497 nm, they correspond to water molecules with two (S2) and four hydrogen bonds (S4), respectively [26]. The water trimer (S2) has been repeatedly connected to the water–protein interaction [106,107,108]. 

Together, these absorbance bands indicate a loss of moisture and consequently a loss of weight in agreement with the reference measurements. Further, they suggest decreased leaf cellulose density and softening and shrinkage of lettuce leaves, consistent with the visual observation of the presence of wilting, especially in the outer leaves, and also that there is an attempt at regulation of defense and survival as the production of vitamin C suggests [109].

In the case of the inner leaves, bands at 1317, 1354, 1397, 1447, 1484, 1515 and 1552 nm were also among the influential variables. The band at 1354 nm related to the first overtone of O-H stretching (OH^―^‧ (H_2_O)_2_) [110] was found to be prominent in fresh bruise detection in apple tissue [111]. The band at 1397 nm is frequently attributed to trapped water [83] and, interestingly, a very close band (1398 nm) also showed high correlation with vitamin C content and shrinkage % of apple slices during hot air drying [97]. An absorption change at 1447 nm can be assigned to the water solvation shell OH^―^‧ (H_2_O)_4,5_ [110], while the one at 1484 nm can be assigned to water molecules with four hydrogen bonds [26]. The 1484 nm band has also been assigned to cellulose [72], but it is more likely that it is actually hydrated cellulose, in other words, water absorbed by cellulose [112,113]. Band 1515 nm is a band of strongly bound water, even though it falls slightly outside the range of C12 WAMACs [26]. The absorbance bands within C12 and closely located bands have been, in numerous works, assigned to the structural water related to preservation, viability and damage [24,85,114,115,116,117], especially when they appeared in combination with trapped water [85,115,118,119]. The last band, 1552 nm (3221 cm^−1^), can be assigned to the hydrogen bonds formed in one crystalline allomorph of cellulose (triclinic I_α_ phase) [120,121,122]. Three bands that are very close or coincide with the ones found in this study, namely 1510, 1484 and 1562 nm, were reported to represent absorbances related to “starch” damage [123]. Starch and cellulose are very similar polymers and, considering the similarities with what was already observed, there are very strong indications of damage to cellulose structures, but this was shown through the changes in the associated water. 

For the outer leaves, there were additional important absorbance bands at 1323, 1348 and 1559 nm. The first two were related mainly to proton hydration [26] while the latter one was the typical bond vibration of the O-H stretch in crystalline cellulose [72,124]. Besides these, prominent bands were found at 1404, 1441, 1453 and 1583 nm. These bands can be attributed to free water molecules [26], water dimers [26], bulk water [24] and highly organized water structures expected around hydrated macromolecules [69], respectively. 

One more important conclusion can be reached about the difference between the inner and outer leaves. In the case of the outer leaves, the best results were obtained using the detrending method, which eliminates variation in the baseline shift caused by the scatter and the differences in leaf tissue structure [34]. The need to use this pre-processing shows that outer leaves suffered physical changes during storage that were not present in the inner leaves. 

#### 3.4.4. Partial Least Squares (PLS) Regression Modeling of Storage Time

PLS regression analysis was performed on the data of inner and outer lettuce leaves to determine the accuracy with which the spectra can be used to predict the storage time (as a dependent variable) in the wavelength range under investigation. The algorithm used in the analyses allowed the exclusion of outliers in the predictive models. As detailed in “Materials and Methods” but in these evaluations as well, the model calibration and validation were conducted by leaving one lettuce head out. The best possible prediction results, namely the estimated storage time based on NIR spectra compared with the actual storage time, are shown in Figure 11. 

Overall, the constructed models demonstrated a good linear relationship between the spectral data and storage time. Comparing the model performances, it was observed that the prediction was slightly better for the outer leaves relative to the inner ones (Figure 11b). This was somewhat expected, as the difference spectra already suggested that, put simply, the inner leaves are protected; therefore, there is some “lag” in response to the environmental changes with time, compared to the outer leaves (Figure 11a). In model validation, the coefficient of determination and RMSE values were R^2^=0.80 and 0.77 days for inner leaves, and R^2^=0.86 and 0.66 days for outer leaves, respectively. This means that by using the NIR spectra we could predict the time the lettuce was in storage for with an error of approximately half a day. This error can probably be reduced significantly if the model is built on a larger sample size and an optimized spectral pre-treatment is applied that can diminish the effect of physical differences between lettuce heads. 

To examine which variables provide the most information about the time spent in cold storage, the PLS regression vectors of the respective PLS regression models for inner and outer lettuce leaves are presented in Figure 12.

The majority of the identified bands in regression vectors coincide and their signs are identical, but there is a significant difference in the overall magnitude. 

The peaks found in the wavelength ranges of 1360–1397 nm and 1503–1589 nm proved to be important contributors to the development of both storage time models. The further inspection and comparison of the regression vectors facilitated the identification of wavelengths at which water conformational changes were detectable and specific for inner or outer lettuce leaves. Absorbance bands at 1329, 1336, 1348, 1459, 1472, 1478, 1484 and 1521 nm turned out to be specific bands for inner leaves, while 1317, 1342, 1453, 1490, 1515 and 1559 nm were for outer leaves. For both predictive models, it was found that the best fit was obtained using detrending pre-treatment.

The pre-processing step allowed the reduction or elimination of the baseline caused by light scattering [34], due to the differences in leaf morphology or even caused by physical damage, during wilting of varying intensity depending on the position of the leaves on the plant. These exploratory analyses of influential variables also confirm that the wavelength range investigated with aquaphotomics proved to be particularly sensitive and reliable for tracking the effects of cold storage, owing to the fact that molecular changes taking place are discernable with great precision. 

#### 3.4.5. Aquagrams 

To further investigate the changes in lettuce during the six-day-long storage and present the findings in a succinct manner, all the prominent absorption bands discovered in the difference spectra, PCA loadings, LDA wavelength contributions and PLS regression vectors are summarily presented in Table 3 and Table 4 separately for inner and outer leaves. 

The bands that consistently occurred during the analyses can now be considered as WAMACs—the locations of NIR spectra, in other words, coordinates at which the changes in the absorbance of leaves will provide information about the state of lettuce. The following wavelengths were selected to be used as WAMACs for the visualization of the spectral pattern of leaves in the aquagrams: 1348, 1360, 1373, 1385, 1391, 1410, 1416, 1422, 1441, 1447, 1453, 1466, 1472, 1490, 1503, 1515, 1521, 1534, 1571 nm. The absorbance values at these WAMACs were used to calculate aquagrams. The aquagrams present the average spectral differences on each day of storage compared to the first measurement day, which when visualized together form the water spectral patterns (WASPs) of lettuce leaves (Figure 13). WASPs can be thought of as multidimensional, integrative biomarkers describing the state of the lettuce leaves and the dynamics of their changes over time spent in cold storage. 

The aquagrams clearly showed that the inner and outer leaves responded differently to the six-day-long cold storage. Compared to the first day, both inner and outer leaves showed lower but varying absorbance at 1348 nm (C1); however, the first significant differences occurred in the area of weakly bound water. In the case of inner leaves, refrigeration caused a decrease in absorbance at 1360 (C2), 1373 (C3), 1385 (C4) and 1391 nm, which did not change significantly after the second day. The outer leaves behaved similarly to the inner leaves on the second day of storage, but thereafter a steady increase could be observed as the storage time progressed. The absorption at these bands is related mainly to proton hydration, ion hydration and trapped water. The increase in absorbance at 1360, 1373 and 1385 nm (water vapor bands [24]) indicates the loss of moisture in the outer leaves, while the increase at 1391 nm signals dehydration due to damage and water expulsion from the cells due to the rupture of cell walls [26,125]. 

The absorbance at wavelengths 1410 nm (C5), 1416 nm and 1422 nm (C6) and 1441 nm (C7), 1447 nm and 1453 nm (C8) showed dynamic change during the storage time but, commonly for both inner and outer leaves, it decreased during storage. The loss of free water molecules (1410 nm) can be related to the loss of juiciness [24]. Both 1416 and 1422 nm bands are hydration water directly associated with the structural elements of leaves, mainly amorphous cellulose as described before, and the decrease in absorbance signals internal damage, which is probably indirectly related to the production of vitamin C as a defense response and may indicate the wilting of the leaves due to this collapse of structures. The density and shrinkage found to be related to this WAMAC in this case indicates a loss of rigidity of structures and turgidity of plant cells, and that the cytoplasm in the plant cells is “less liquid” and more amorphous. The decrease in absorbance at 1441 nm and 1447 nm further describes the loss of water interacting with cellulose, either the water molecules directly associated with glucose molecules in the cellulose, or the water molecules in its solvation shell or envelope. This further supports the loss of turgor in plant cells and shrinking of the cytoplasm from the cell walls. 

Between 1441 and 1490 nm, the inner leaves showed a higher absorbance on the second day compared to the first day’s results, which then decreased sharply and did not really change in the subsequent days; however, in the same spectral region, the outer leaves showed a decrease, especially in the last two storage days. The decrease in absorbance of leaves in this region is associated with the loss of small hydrogen-bonded water clusters with two, three and four hydrogen bonds. The decrease in absorbance in this region shows the loss of water interacting with proteins, hydrated cellulose and other polysaccharides, and increased crystallinity of the remaining water. The WASPs that have high absorbance in this region are usually associated with still crispy and juicy flesh [21]; therefore, during storage the lettuces probably lost this crispiness and juiciness. The low absorbance in the region 1344–1382 nm and high absorbance at 1410–1492 nm was found to be associated with firm, juicy and crispy sensory profiles in apples [21]. In the case of lettuce here, there is no doubt that juiciness is lost in both the external and inner zones of the lettuce head, but the remaining high values of absorbance in the region 1441–1490 nm in the inner leaves testify to preserved firmness compared to the outer leaves. 

A clear increase in light absorbance can be observed in the region between 1503 and 1571 nm. All the absorbance bands located within this range are indicative of strongly molecularly bound crystalline water. The increased absorbance at this region, for all of the associated bands, is a clear consequence of the loss of liquid water, which led to the shrinking of plant cells and led them to collapse in upon themselves, with the only remaining water species being those who were bound to the structural elements of plant tissues. 

In summary, it can be concluded that WASPs of the inner and outer leaves are quite similar, telling the same story of what is happening with the leaves during 6 days of cold storage, with enough precision to tell the difference between internal quality and to show that some of it is still preserved in the inner leaves due to the protection of their location. The bands presented here as WAMACs, for the first time identified and used for the description of the lettuce leaves during storage, are similar and consistent with findings of other researchers who worked with similar systems and similar research objectives (Table 5). 

Our research, however, contributed several more absorbance bands (1416, 1447, 1503, 1521, 1534 and 1571 nm) which is probably due to the specific structure of the investigated system (lettuce) and also due to progress in the understanding of the water structures absorbing at the 1520 nm wavelength. The agreement with existing studies testifies to the universality of the presented aquaphotomics method for the description of changes during storage and shows that it can be used successfully for the monitoring of various vegetable and fruit systems for the evaluation of their quality during storage and shelf life.

## 4. Conclusions

This study was conducted with the objective of developing a non-destructive, aquaphotomics-based methodology for monitoring changes during the cold storage of lettuce utilizing a water spectral pattern (WASP) of lettuce leaves as a multidimensional biomarker that describes their state and dynamics over the time spent in storage.

An aquaphotomics analysis of the lettuce spectra was conducted in the first overtone region of water (1300–1600 nm) and included the exploration of difference spectra, exploratory analysis PCA, discriminating analysis LDA and regression analysis PLSR, using time as a dependent variable. The spectra of the inner and outer leaves of lettuce showed very different spectral profiles, because the location of the leaves dictated the level of exposure to the environmental impact. The analyses were therefore performed separately for the spectra of inner and outer leaves. The results of these analyses consistently showed that the changes in both inner and outer leaves of lettuce can be primarily determined and related to the alterations of water molecular structures in the leaves. Supervised statistical methods have allowed very accurate classification and the prediction of time spent in cold storage based on the spectral data. The systematization of influential variables in the developed models allowed for the identification of the water absorbance bands at which the absorbance changes during storage are related to the state of the lettuce. These bands were adopted as WAMACs, and further used as coordinates to display the WASPs of lettuces during storage. The defined WASPs were used as multidimensional biomarkers that described the state of lettuces during the storage period. 

The comparison of WASPs of lettuce leaves on different days during cold storage showed that, compared to the first day of storage, the leaves lost the free and weakly bound water and there was a rupture of cell walls and damage, leading to the altered state of cytoplasm in plant cells that could be described as more amorphous. The defense response against the damage, shown by a production of vitamin C, was also observed. Further, the lettuce leaves showed shrinkage, decreased density and collapse of the internal tissue structures with tissue water organized predominantly in the bound state, bonded to the structural elements such as cellulose and other polymers. This information was entirely reflected in the water spectral patterns—WASPs of the inner and outer leaves of lettuce heads. 

It can be concluded that the water spectral pattern provides a fingerprint-like universal analytical image about the given samples that can be interpreted from a physiological state and quality point of view as well. The findings of this study provide a basis for widening the horizons of aquaphotomics investigations in the field of post-harvest monitoring and quality control. 

## Figures and Tables

**Figure 1 foods-12-00258-f001:**
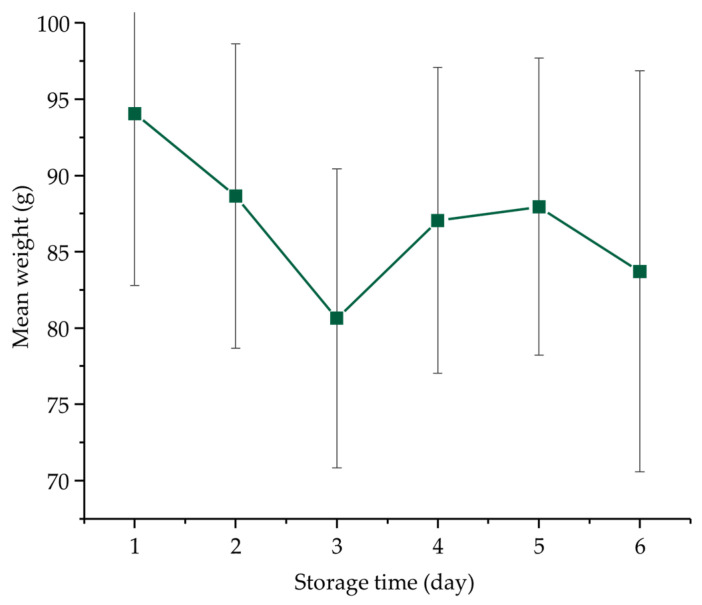
Temporal changes in weight of lettuce heads during the six days of cold storage. Data are presented as mean ± standard error.

**Figure 2 foods-12-00258-f002:**
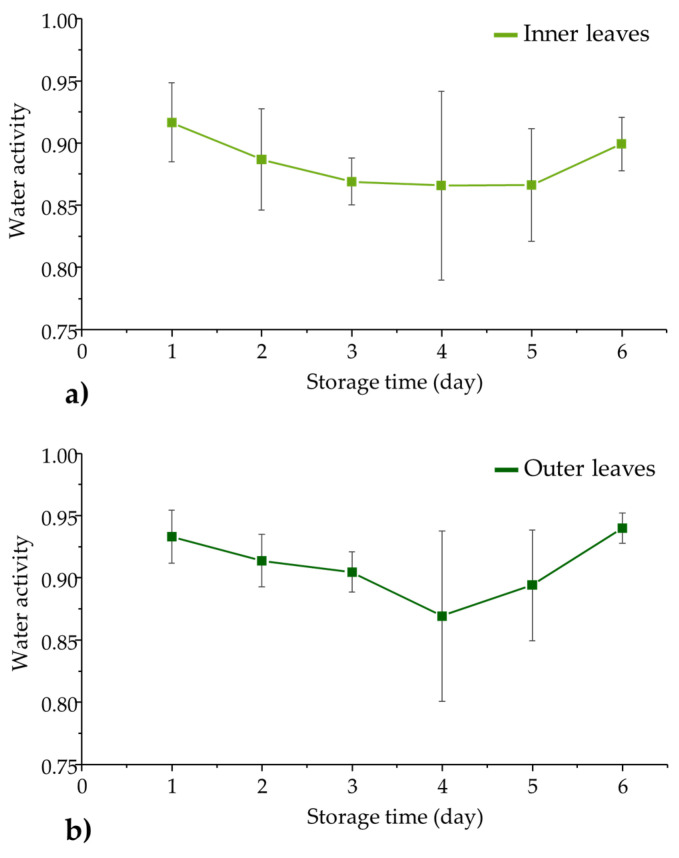
Temporal changes in water activity of lettuce during the six days of cold storage for (**a**) inner leaves and (**b**) outer leaves. Data are presented as mean ± standard error.

**Figure 3 foods-12-00258-f003:**
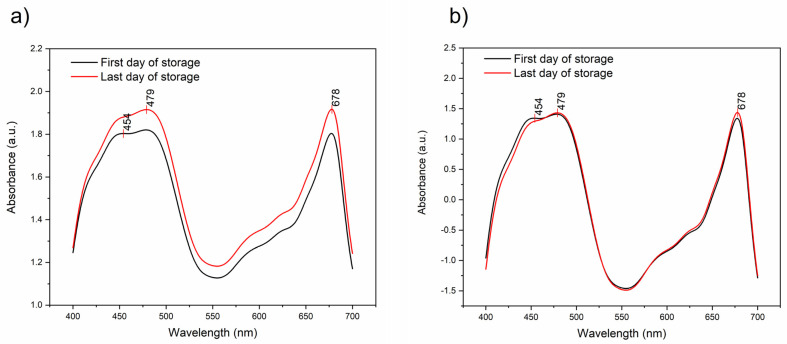
The changes in the visible part of the spectra. (**a**) Raw spectra: comparison of average spectra of lettuce leaves acquired on the first and last day of storage; (**b**) SNV-transformed spectra: comparison of average spectra of lettuce leaves acquired on the first and last day of storage.

**Figure 4 foods-12-00258-f004:**
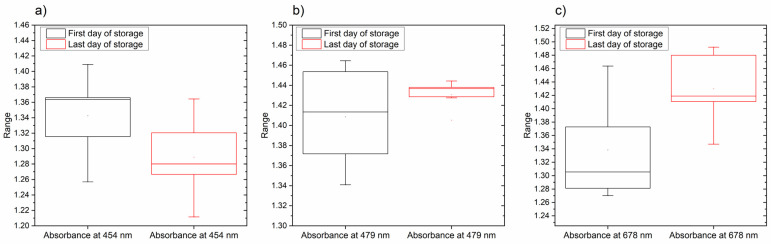
Comparison of values of absorbance at absorbance bands corresponding to the identified main pigments in lettuce leaves—454 nm (β-carotene), 479 nm (Chl b) and 678 nm (Chl a): (**a**) values of absorbance at 479 nm; (**b**) values of absorbance at 454 nm; (**c**) values of absorbance at 678 nm. The absorbance values are averages from the spectra of lettuce leaves acquired on the first and last day of storage.

**Figure 5 foods-12-00258-f005:**
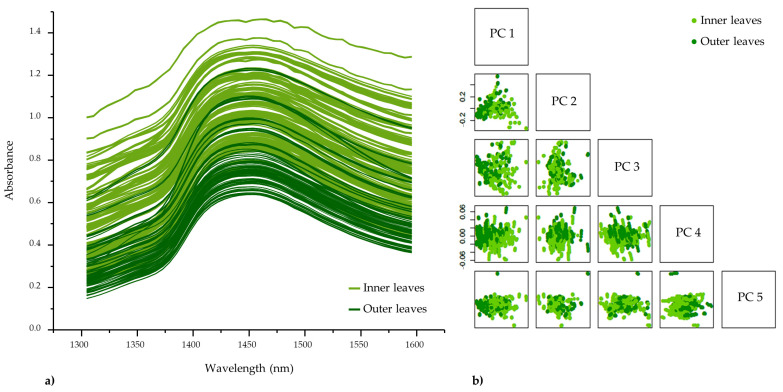
(**a**) Raw spectra of lettuce leaves (N = 900); (**b**) PCA score plot on raw data colored according to “lettuce leaves” (N = 900).

**Figure 6 foods-12-00258-f006:**
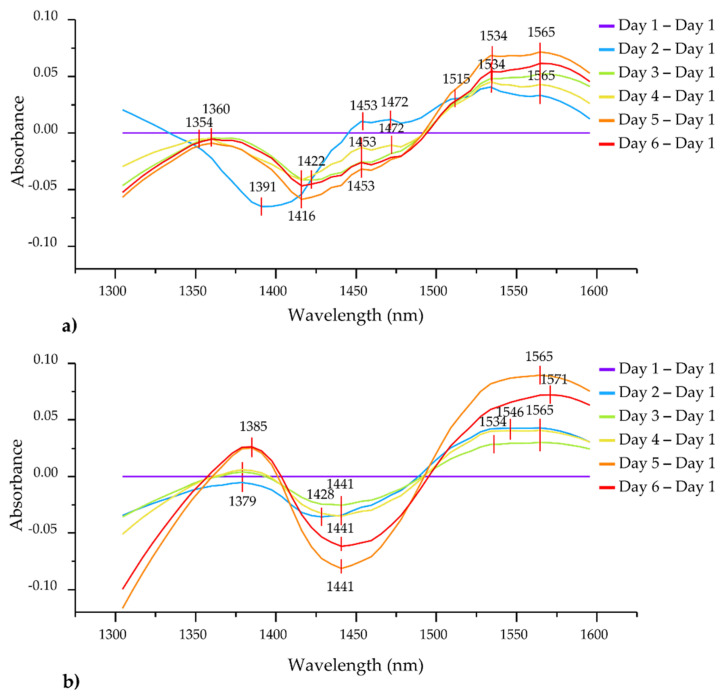
Smoothed, SNV-treated and averaged difference spectra of inner leaves (N = 6) (**a**) and outer leaves (N = 6) colored according to the storage duration (**b**).

**Figure 7 foods-12-00258-f007:**
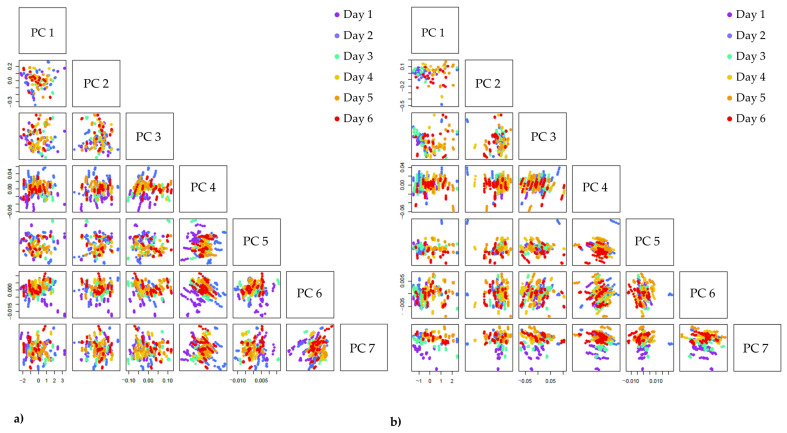
The score plots of PCA analyses performed using the raw data of (**a**) inner leaves (N = 450) and (**b**) outer leaves (N = 450). The scores are colored according to the number of days spent in cold storage.

**Figure 8 foods-12-00258-f008:**
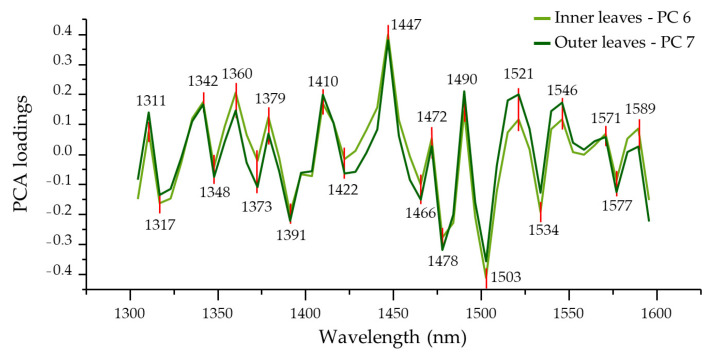
Selected PCA loadings of inner leaves (PC 6) and outer leaves (PC 7).

**Figure 9 foods-12-00258-f009:**
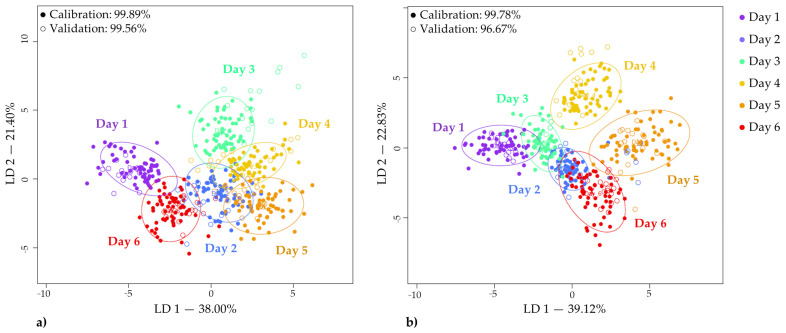
PCA-LDA classification models when “storage time” was the class variable (**a**) on the data of inner leaves (raw spectra, N = 450, NrPCs = 24), and (**b**) on the data of outer leaves (deTr-treated spectra, N = 450, NrPCs = 30).

**Figure 10 foods-12-00258-f010:**
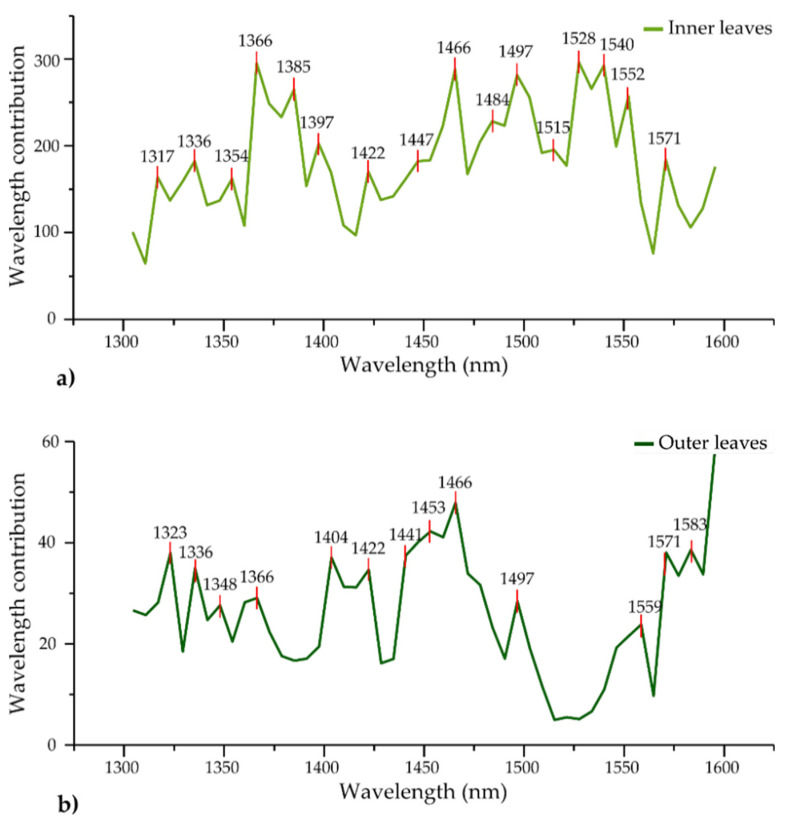
Wavelength contributions to PCA-LDA classification model when “storage time” was the class variable (**a**) on the data of inner leaves (raw spectra, N = 450, NrPCs = 24), and (**b**) on the data of outer leaves (deTr-treated spectra, N = 450, NrPCs = 30).

**Figure 11 foods-12-00258-f011:**
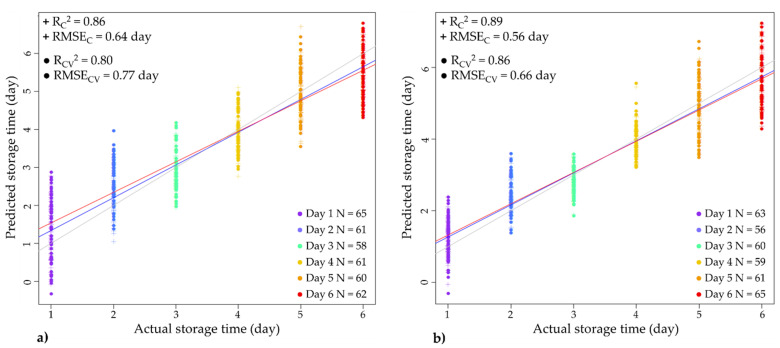
PLSR models for storage time prediction. (**a**) Agreement between the actual and predicted storage time (days) using the spectral data of inner leaves (deTr-treated spectra, N = 367, NrLV = 9) and (**b**) agreement between the actual and predicted storage time (days) using the spectral data of outer leaves (deTr-treated spectra, N = 364, NrLV = 9).

**Figure 12 foods-12-00258-f012:**
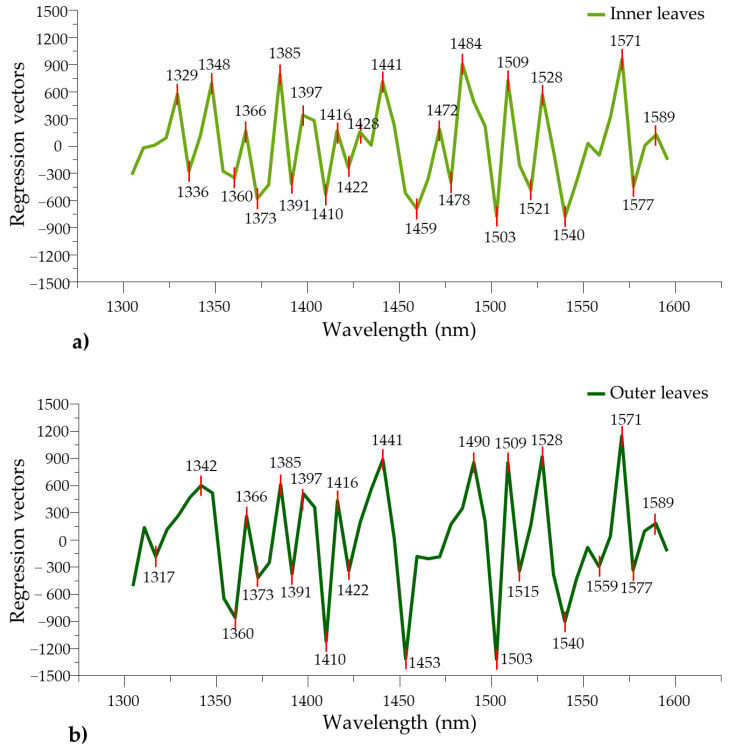
PLS regression vectors for storage time prediction. (**a**) Data of inner leaves (deTr-treated spectra, N = 367, NrLV = 9) and (**b**) data of outer leaves (deTr-treated spectra, N = 364, NrLV = 9).

**Figure 13 foods-12-00258-f013:**
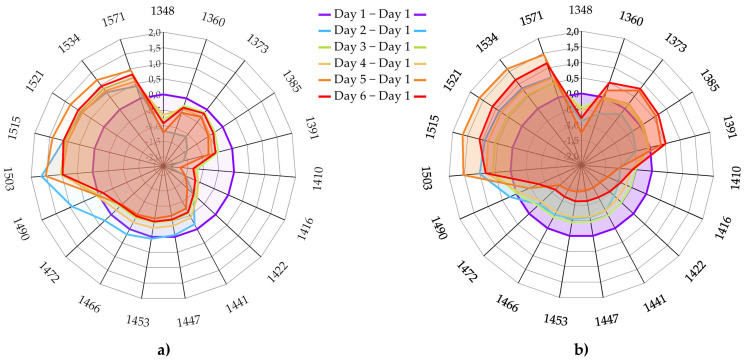
Aquagrams of lettuce leaves calculated for each day of storage for (**a**) inner leaves (N = 6), and (**b**) outer lettuce leaves (N = 6). The aquagrams are calculated using pre-treated spectra (sgol-2-17-0 + SNV + averaging for all spectral replicates and consecutive replicates) and subtraction of the aquagram of the leaves for the first day of storage.

**Table 1 foods-12-00258-t001:** PCA-LDA classification results on the data of inner leaves when “storage time” was the class variable (raw spectra, N = 450, NrPCs = 24).

Accuracy	(%)	Day 1	Day 2	Day 3	Day 4	Day 5	Day 6	Correct Classification
Calibration	Day 1	100	0	0	0	0	0	99.89%
	Day 2	0	99.33	0	0	0	0	
	Day 3	0	0	100	0	0	0	
	Day 4	0	0	0	100	0	0	
	Day 5	0	0	0	0	100	0	
	Day 6	0	0.67	0	0	0	100	
Validation	Day 1	100	0	0	1.33	0	0	99.56%
	Day 2	0	93.33	0	0	2.67	2.67	
	Day 3	0	0	100	0	0	0	
	Day 4	0	1.33	0	98.67	0	0	
	Day 5	0	0	0	0	97.33	1.33	
	Day 6	0	5.33	0	0	0	96	

**Table 2 foods-12-00258-t002:** PCA-LDA classification results on the data of outer leaves when “storage time” was the class variable (deTr-treated spectra, N = 450, NrPCs = 30).

Accuracy	(%)	Day 1	Day 2	Day 3	Day 4	Day 5	Day 6	Correct Classification
Calibration	Day 1	98.67	0	0	0	0	0	99.78%
	Day 2	0	100	0	0	0	0	
	Day 3	1.33	0	100	0	0	0	
	Day 4	0	0	0	100	0	0	
	Day 5	0	0	0	0	100	0	
	Day 6	0	0	0	0	0	100	
Validation	Day 1	93.33	0	0	0	0	0	96.67%
	Day 2	1.33	89.33	0	0	0	1.33	
	Day 3	5.33	0	100	0	0	0	
	Day 4	0	0	0	100	0	0	
	Day 5	0	5.33	0	0	98.67	0	
	Day 6	0	5.33	0	0	1.33	98.67	

**Table 3 foods-12-00258-t003:** Prominent wavelengths for statistical modeling on the data of inner leaves.

WAMACs		C1		C2	C3	C4		C5	C6	C7	C8	C9	C10	C11	C12			
Wavelength Range	1310–1334	1336–1348	1350–1358	1360–1366	1370–1376	1380–1388	1390–1396	1398–1418	1421–1430	1432–1444	1448–1454	1458–1468	1472–1482	1482–1496	1506–1516	1518–1538	1540–1559	1560–1590
**Difference spectra**																		
Day 2–Day 1							**1391**				**1453**		**1472**		**1515**	**1534**		1565
Day 3–Day 1				**1360**					**1422**		**1453**					**1534**		1565
Day 4–Day 1			1354					**1416**			**1453**		**1472**			**1534**		1565
Day 5–Day 1				**1360**				**1416**			**1453**					**1534**		1565
Day 6–Day 1				**1360**				**1416**			**1453**					**1534**		1565
**PCA loadings**																		
PC 1					**1373**													
PC 2											**1447**							
PC 3						**1385**					**1453**							
PC 4	1311						**1391**				**1447**		**1472**	**1490**		**1521**	1546	**1571**, 1589
PC 5		1336, **1348**			**1373**			1397	1428			1459	1478	**1490**		**1521, 1534**		1577
PC 6	1311, 1317	1342, **1348**		**1360**	**1373**	1379	**1391**	**1410**	**1422**		**1447**	**1466**	**1472**, 1478	**1490**	**1503**	**1521, 1534**	1546	**1571**, 1577, 1589
PC 7			1354					**1410**					1478					
**LDA wavelength contribution**	1317	1336	1354	1366				1397	**1422**		**1447**	**1466**		1484, 1497	**1515**	1528	1540, 1552	**1571**
**PLS regression vectors**	1329	1336, 1348		**1360**, 1366	**1373**	**1385**	**1391**	1397, **1410**	**1422**, 1428	**1441**		1459	**1472**, 1478	1484	**1503**, 1509	**1521**, 1528	1540	**1571**, 1577, 1589

**Table 4 foods-12-00258-t004:** Prominent wavelengths for statistical modeling on the data of outer leaves.

WAMACs		C1		C2	C3	C4		C5	C6	C7	C8	C9	C10	C11	C12			
Wavelength Range	1310–1334	1336–1348	1350–1358	1360–1366	1370–1376	1380–1388	1390–1396	1398–1418	1421–1430	1432–1444	1448–1454	1458–1468	1472–1482	1482–1496	1506– 1516	1518–1538	1540–1559	1560–1590
**Difference spectra**																		
Day 2–Day 1						1379			1428								1546	1565
Day 3–Day 1						1379				**1441**						**1534**		1565
Day 4–Day 1						1379				**1441**							1546	1565
Day 5–Day 1						**1385**				**1441**								1565
Day 6–Day 1						**1385**				**1441**								**1571**
**PCA loadings**																		
PC 1								**1416**							**1515**			
PC 2											**1447**							
PC 3						**1385**					**1447**							
PC 4							**1391**							**1490**	**1515**			
PC 5				**1360**				1404				1459				**1521**		
PC 6				**1360**		1379		**1416**					**1472**	**1490**	**1515**			
PC 7	1311, 1317	1342, **1348**		**1360**	**1373**		**1391**	**1410**	**1422**		**1447**	**1466**	**1472**, 1478	**1490**	**1503**	**1521, 1534**	1546	**1571**, 1577, 1589
**LDA wavelength contribution**	1323	1336, **1348**		1366				1404	**1422**	**1441**	**1453**	**1466**		**1497**			1559	**1571**, 1583
**PLS regression vectors**	1317	1342		**1360**, 1366	**1373**	**1385**	**1391**	1397, **1410**, **1416**	**1422**, 1428	**1441**	**1453**			**1490**	**1503**, 1509, **1515**	1528	1540, 1559	**1571**, 1577, 1589

**Table 5 foods-12-00258-t005:** Agreement between the WAMACs found for description of changes during storage.

Monitoring Storage of Lettuce (This Study)	1348	1360	1373	1385	1391	1410	1422	1441	1453	1466	1472	1490	1515
Mung bean germination [126]	1343	1364	1374	1383		1411	1426	1441	1453	1462	1477	1489	1513
Monitoring pineapple slice solar dehydration [118]	1342	1366	1373			1410	1428	1441	1453	1459	1478	1490	1515
Monitoring rice germ storage [24]	1343	1364	1375	1382	1392	1410	1425	1436	1455		1474	1492	1518
Storage monitoring of rocket salad [22]	1342	1366	1373	1385		1416	1428	1441	1453	1466	1478	1490	1509
Studying the influence of packaging and coating materials during storage of winter melons [127]	1344	1364	1372	1382	1398	1410		1438 1444		1464	1474	1492	1518
Studying apple sensory texture of stored apples [21]	1344	1364	1372	1382	1398	1410		1438 1444		1464	1474	1492	1518

## Data Availability

The data that support the findings of this study are available on request from the corresponding author (RT).

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
