# Peer review of "Aquaphotomics Monitoring of Lettuce Freshness during Cold Storage"

_foods, 2023, doi:10.3390/foods12020258_

Round 1

Reviewer 1 Report

I reviewed the manuscript entitled, Aquaphotomics Monitoring of Lettuce Freshness During Cold Storage. The manuscript is well written and contributes to the field. In my opinion, authors should consider the suggestions below.

Abstract: Please introduce the background of the study

Lines 113-116: please revise the last objective. Why is the objective in future tense? Is it the author's research or recommendation?

What is storage time (3) in line 116

Please provide the space between sold [28] and On

Provide the ref for 2.1.2. Water Activity Measurements

Please provide the ref for 2.1.3. Evaluation of Color Changes

Please provide the ref for 2.1.4. NIR spectral acquisition

Detailed methodology on PCA, LDA and PLSR should be needed with references

Section 2.3.2. Aquaphotomics multivariate data analysis is not methodology. It is background information. Please provide methodology for the statistical methods used

Lines 238-250: it's not results and just a background information

Figures 1 and 2. Why is there so much standard deviation?

Section 3.2. Water Activity Changes. Please revise it as “3.2. Water Activity”

Improve the quality of Figures 3 and 4

Line 330: Is it Figure 1 or Figure 4?

Line 384: Figure number? It seems there is a problem in Figure numbers. I suggest to revise

Line 461: Figure 7 but PCA is with Figure 4. What is correct?

Classification and validation findings are interesting and in line with literature.

Authors should revise the conclusions to reflect the conducted study and recommendations

References are not according to the journal guidelines

Author Response

Reviewer #1: General remarks

I reviewed the manuscript entitled, Aquaphotomics Monitoring of Lettuce Freshness During Cold Storage. The manuscript is well written and contributes to the field. In my opinion, authors should consider the suggestions below.

Authors’ response

We are very grateful for your endorsement of this work, and we carefully went over the Manuscript to make all the corrections you indicated.  All the revisions in the Manuscript are highlighted in yellow for easy tracking. Please find all the specific comments and our point-to-point replies below, where we describe what revision was performed and where it can be found in the revised Manuscript.

We hope that we successfully addressed all the issues and that paper can now be accepted for publication.  Thank you for investing your time and expertise to help us improve this Manuscript.

Authors

--------------------------------------â‚°-------------------------------------------------

Reviewer #1: Specific comments

Comment 1

Abstract: Please introduce the background of the study

Response 1

Thank you for this suggestion. The background is now provided in the abstract. Please find revision at L14-L17.

Comment 2

Lines 113-116: please revise the last objective. Why is the objective in future tense? Is it the author's research or recommendation?

Response 2

Thank you for this advice. The objectives are now revised to be in the correct tense. We have actually split the formerly last objective into 2 objectives to simplify the sentences. We revised the paragraph to show that the listed objectives were the objectives we wanted to achieve with the present work, not a recommendation for someone else. Please find the revision at L118-L126.

Comment 3

What is storage time (3) in line 116

Response 3

With the term “storage time” we meant the time the lettuce spent in cold storage. We have revised this expression. Please see the revision at L126.

Comment 4

Please provide the space between sold [28] and On

Response 4

Thank you for noticing this! The space is added. Please see revision at L144.

Comment 5

Provide the ref for 2.1.2. Water Activity Measurements

Response 5

We provided the revision and also added 3 new references that can be used as sources to enquire about what “water activity” means, and for the methods used to measure it. Please find the revised lines L147 and L148-L151.

Comment 6

Please provide the ref for 2.1.3. Evaluation of Color Changes

Response 6

We provided new opening sentence for the 2.1.3 section and added a reference as requested. Please find the revision at L157-158.

Comment 7

Please provide the ref for 2.1.4. NIR spectral acquisition

Response 7

We provided new opening sentence for the 2.1.4 section and added a reference as requested. Please find the revision at L168-170.

Comment 8

Detailed methodology on PCA, LDA and PLSR should be needed with references

Response 8

We have revised the Data analysis section according to this advice. We also noticed that we missed to provide the reference in the section 2.3.1 when Standard normal variate transformation was first mentioned and we corrected it by supplying the reference, please see the revision at L185.

Detailed methodology is now provided in the section 2.3.2. The PCA, LDA and PLSR analyses were described and adequate references supplied. Please find the revisions at L206 -L215 for PCA, L216-L220 for LDA, and for PLSR at L235-240.

Comment 9

Section 2.3.2. Aquaphotomics multivariate data analysis is not methodology. It is background information. Please provide methodology for the statistical methods used

Response 9

The analysis was performed according to the guidelines given in our previous publication, and this information is added at L186 and the following reference is supplied. 

Tsenkova, R., Munćan, J., Pollner, B., & Kovacs, Z. (2018). Essentials of aquaphotomics and its chemometrics approaches. Frontiers in chemistry6, 363.

Comment 10

Lines 238-250: it's not results and just a background information

Response 10

We have decided to remove the lines 246-252 as they are indeed background information and do not belong to the Results section. However, the lines 254-258 contained important background information that we decided to move to the Introduction as it is relevant. Please find the revision at L108-L112 in the Introduction, and also in the beginning of the Results section, where the paragraph is actually now removed between the heading 3. Results and Discussion and 3.1. Weight change (L271-L272).

Comment 11

Figures 1 and 2. Why is there so much standard deviation?

Response 11

The reason for large standard deviation seems to originate from differences between biological replicates. From our experience with a variety of not only leafy, but other vegetables, large deviation in weight changes and also water activity between samples is a common occurrence. We checked the available literature, and it seems that the other researchers also reported similarly large standard deviation in the weight of lettuce. For example, the following two publications report such results:

1)       Gent, M. P. (2012). Composition of hydroponic lettuce: effect of time of day, plant size, and season. Journal of the Science of Food and Agriculture, 92(3), 542-550 (for example, average weight was found to be 295 ± 49g 63 days after germination)

2)       McLain, J., Castle, S., Holmes, G., & Creamer, R. (1998). Physiochemical characterization and field assessment of lettuce chlorosis virus. Plant disease, 82(11), 1248-1252 (for example, one of the 4 varieties of lettuce had 873.2±109g head weight)

Therefore, it can be expected that the change in day-to-day weight will also show high variability between different lettuce heads (biological replicates). We think the same reason is valid for Figure 2 and water activity measurements.

Regarding this comment we did not provide any revision in the manuscript, but we highlighted the sentence at L274-L277, where the explanation about the large standard deviation was provided in the original manuscript and attributed to naturally high variability of the sample replicates.

Comment 12

Sect-ion 3.2. Water Activity Changes. Please revise it as “3.2. Water Activity”

Response 12

The title is revised as advised, please find the revision at L299.

Comment 13

Improve the quality of Figures 3 and 4

Response 13

Thank you for this advice. We regret that images were of low quality. In the revised manuscript newly provided figures 3 and 4 are created with 600 dpi resolution. Please find the new figures at L332 and L349.

Comment 14

Line 330: Is it Figure 1 or Figure 4?

Response 14

This was a mistake. We corrected the figure number to Figure 4. Please find the revision at L350.

Comment 15

Line 384: Figure number? It seems there is a problem in Figure numbers. I suggest to revise

Response 15

Yes, unfortunately we did not see this. Thank you for your help. The figure numbers for figures 3 to 13 are now corrected. Please find the corrected numbering of figures at L332, L350, L404, L413, L485, L494, L544, L559, L630, L660 and L691.

Comment 16

Line 461: Figure 7 but PCA is with Figure 4. What is correct?

Response 16

This was a mistake. Thank you for correcting us. The figure number is corrected (L485) and it matches with the “Figure7” in the text at L481. These mistakes are now also corrected in the entire manuscript.

Comment 17

Classification and validation findings are interesting and in line with literature.

Response 17

Thank you for checking this and for the endorsement of our work.

Comment 18

Authors should revise the conclusions to reflect the conducted study and recommendations

Response 18

Thank you for the advice. We have revised the section “4. Conclusions” according to the instructions given by the reviewer, please find the revision at L759-L786.

Comment 19

References are not according to the journal guidelines

Response 19

Thank you for noticing this. We have corrected many references that lacked the critical information, removed 3 instances of duplicates, added missing doi numbers and the references are now arranged using the Foods style according to the Journal instructions. Please find the revised Reference list at L814-L1122.

Authors are very grateful for the comments of Reviewer 1.

Reviewer 2 Report

The experimental material comprised of ten heads of lettuce purchased in one supermarket. The material should be more diverse.

There is no detailed information on the variety, growing conditions, etc.

Considering such limited material, it is difficult to conclude the importance of the results.

Research should be carried out for a larger number of varieties with characterized growing conditions.

Author Response

Reviewer #2: General remarks

The experimental material comprised of ten heads of lettuce purchased in one supermarket. The material should be more diverse.

There is no detailed information on the variety, growing conditions, etc.

Considering such limited material, it is difficult to conclude the importance of the results.

Research should be carried out for a larger number of varieties with characterized growing conditions.

Authors’ response

Dear Reviewer,

We are very grateful for your investment of time and expertise to review our Manuscript, and we appreciate your critical opinions.

We do agree that the number of samples is small, but the manuscript we submitted and this research performed on lettuce, is just one little part of a large study we performed and which included a large variety of fruits, including different cultivars  (grapes, strawberries, avocado, banana etc.) and vegetables (lettuce, asparagus, tomato etc.), grain (rice, quinoa, wheat, etc.), as well as a variety of food products (butter, bread, coffee powder, chewing gum, crackers etc.). Our main research interest in this project was to examine the behavior of fresh food produces and food products during storage (stored at different levels of water activity ranging from 0.2 to 0.99 aw) using aquaphotomics, understand what water activity means in the terms of water molecular structure, modeling the water activity and also exploring the effect of different storage conditions on preservation. While acquiring the spectra of all these products is rapid and easy, measuring the water activity requires much more time. Hence, to be able to test if our method is applicable for a wide variety of food products, we had to work with smaller number of samples in some cases (such as this paper).

At this point, the analysis of all the experimental data we acquired is finished and we had to think about the strategy of presenting our findings. We have chosen as a best option to present our results for each food type separately and then show the whole story with a review paper where we can summarize the findings from all these papers. The submitted lettuce manuscript is the third we submitted for publication (the first one was about the rice germ “Analysing the water spectral pattern by near-infrared spectroscopy and chemometrics as a dynamic multidimensional biomarker in preservation: rice germ storage monitoring” https://www.sciencedirect.com/science/article/pii/S1386142521009732 , and the second one is about strawberries “Aquaphotomics monitoring of strawberry fruit during cold storage – A comparison of two cooling systems” https://www.frontiersin.org/articles/10.3389/fnut.2022.1058173/full).

As we publish one by one, our findings related to different food systems, the full picture will emerge, presenting the water spectral pattern as a novel multidimensional, integrative marker that single-handedly provides information about the state of freshness, water activity, mechanical, textural and other properties of interest.    

So, based on this one paper, we very well understand that any reviewer or any reader may see our sample size as small, lacking in diversity and may question the importance of our findings. However, it is only a piece of a large body of evidence we collected, and the results we presented in the manuscript are consistent and in agreement with what we observed in our other works – that the water spectral pattern can be used to describe the state of the food system (fresh produce of product), which is something you can now also see in the papers we already published.

In the case of lettuce, we were interested in simulating the typical behavior of a consumer, who will go to the store, buy a lettuce, remove the package and store it in the fridge. We wanted to see if we can track the state of lettuce in the typical conditions after the product is bought in the supermarket and stored, so this is the reason why we did not grow the lettuce by ourselves and why the growing conditions are not supplied. We were not interested in the lettuce per se, their varieties and so on.  

We hope that this response provides better explanation about our main motivation and objective of research, and we hope that in light of this new information reviewer can better understand the importance of this work for us and why we would like it to be published as it is.  Thank you for reading and reviewing our paper, we appreciate your willingness to provide the time and expertise to the help your peers in their scientific endeavors.    

Authors

---------------------------------------------â‚°-----------------------------------------

Reviewer 3 Report

This MS evaluated the freshness changes during cold storage of lettuce through aquaphotomics approach. This study's results indicated that changes in cold storage lettuces are associated with alterations in water molecular structures in leaves. This MS should be proper for Foods, while the minor comments should be addressed. 1. for the '2.2 Methods' section, the 2.1.1 to 2.1.4 should be 2.2.1 to 2.2.4. 2. The figures listed in the MS are chaotic in order and are not consistent with those mentioned in the main text. 3. I would like to suggest the authors move some figures and tables that are not very important into supplemental files.

Author Response

Reviewer #3: General remarks

This MS evaluated the freshness changes during cold storage of lettuce through aquaphotomics approach. This study's results indicated that changes in cold storage lettuces are associated with alterations in water molecular structures in leaves. This MS should be proper for Foods, while the minor comments should be addressed.

Authors’ response

Dear Reviewer,

We are very grateful for your recommendation of our Manuscript for publication with only minor revisions.  All your specific comments are addressed in the point-to-point replies provided below, where we described what revision was performed and where it can be found in the revised Manuscript. All the revisions in the Manuscript are highlighted in yellow for easy tracking.

Thank you for investing your time to read and review our paper, and for the opportunity to improve its quality.

Authors

----------------------------------------------------â‚°-----------------------------------------------------------

Reviewer #3: Specific comments

Comment 1

For the '2.2 Methods' section, the 2.1.1 to 2.1.4 should be 2.2.1 to 2.2.4.

Response 1

Thank you for noticing this omission! We have corrected the numbering order of the headings and subheadings, please find the revisions at L139, L146, L156 and L167.

Comment 2

The figures listed in the MS are chaotic in order and are not consistent with those mentioned in the main text

Response 2

Thank you for bringing our attention to this, we are very sorry that this slipped unnoticed on our part. The numbering of the figures is corrected and consistent in the revised manuscript. Please find the revised numbering of figures at L332, L350, L404, L413, L485, L494, L544, L559, L630, L660 and L691.

Comment3

I would like to suggest the authors move some figures and tables that are not very important into supplemental files.

Response3

Thank you for this suggestion. If reviewer insists, we can perhaps move the Tables 3, 4 and 5 to Supplementary material. While we do agree that having 13 figures and 5 tables is a lot, in our opinion each contributes to the story of the paper. Since the journal does not limit the number of figures or tables, if possible, we would really prefer to have all the results displayed as they currently are, because we think that they show repeatability and consistency of findings, something on which we very much insist in aquaphotomics. We would really appreciate your understanding.

Authors are very grateful for the comments of Reviewer 3.

Round 2

Reviewer 2 Report

The Authors' explanations of my comments did not bring any improvement to the manuscript.